# Molecular Detection of Urogenital Schistosomiasis in Community Level in Semi-Rural Areas in South-East Gabon

**DOI:** 10.3390/diagnostics15091052

**Published:** 2025-04-22

**Authors:** Lady Charlène Kouna, Sandrine Lydie Oyegue-Liabagui, Chenis Nick Atiga, Chérone Nancy Mbani Mpega Ntigui, Roméo Karl Imboumy-Limoukou, Jean Claude Biteghe BI Essone, Steede Seinnat Ontoua, Diamella Nancy Moukodoum, Alain Prince Okouga, Jean Bernard Lekana-Douki

**Affiliations:** 1Unité d’Evolution, Epidémiologie et Résistances Parasitaires (UNEEREP), Centre Interdisciplinaire de Recherche Médicales de Franceville, Franceville BP 769, Gabon; lyds_ass@yahoo.fr (S.L.O.-L.); atiganick@yahoo.fr (C.N.A.); mpega_mb2@yahoo.fr (C.N.M.M.N.); imboumykarl@gmail.com (R.K.I.-L.); biteghebiteghe@gmail.com (J.C.B.B.E.); ontouaseinnat@gmail.com (S.S.O.); nancydiamella@gmail.com (D.N.M.); alainprince.okouga@gmail.com (A.P.O.); lekana_jb@yahoo.fr (J.B.L.-D.); 2Ecole Doctorale Régionale d’Afrique Centrale en Infectiologie Tropicale, Franceville BP 876, Gabon; 3Département de Biologie, Faculté des Sciences, Université des Sciences et Techniques de Masuku, Franceville BP 901, Gabon; 4Département de Parasitologie Mycologie et de Médecine Tropicale, Université des Science de la Santé, Libreville BP 18231, Gabon

**Keywords:** *Schistosoma haematobium*, urinary, qPCR, urine stick, filtration

## Abstract

Background: Urogenital schistosomiasis is endemic in Gabon. Our study aimed to detect the prevalence of urinary schistosomiasis and to evaluate the diagnostic performance of the qPCR technique compared to microscopy for the detection of *Schistosoma haematobium* at the community level in a semi-rural area. Method: A cross-sectional survey was carried out. Urine samples were examined using Urine TICK test strips, a filtration technique, and qPCR. *Schistosoma haematobium* was detected by targeting the Dra1 gene. Results: The prevalence of urogenital schistosomiasis was determined and the performance of real-time PCR and urine strips was compared with that of urinary filtration. A total of 281 participants were enrolled in the study. The prevalence of urogenital schistosomiasis was increased slightly with the molecular technique (40.9%) compared to microscopy (36.7%), and the hematuria rate with Urine STICK was 33.5%. SAC (5–14 years old), Pre-SAC (>5 years old), and adolescents (15–17 years old) were the most affected group according to, respectively, whatever method was used. qPCR showed good agreement with microscopy, as well as excellent sensitivity (99.03%) and specificity (93.3). There was a good correlation between the number of eggs per 10 mL and the cycle threshold range. Conclusion: These results show the importance of using a combination of diagnostic tools in the surveillance of schistosomiasis, particularly in preschool children, adolescents, women of childbearing age, and chronic and asymptomatic cases.

## 1. Introduction

Human schistosomiasis is one of the most widespread neglected tropical diseases (NTDs) and is a major global public health problem. It is estimated that 700 million people are at risk of infection and 200 million will be infected by schistosomiasis in the future [1]. It is endemic in 78 countries, particularly in sub-Saharan Africa and tropical zones, where the highest infection rates have been observed, due to a lack of adequate hygiene. Urinary and intestinal schistosomiasis are endemic in 54 and 56 countries, respectively. This parasitic epidemic is caused by trematodes of the genus Schistosoma [1].

Urogenital schistosomiasis, widely distributed throughout Africa and the Middle East, is caused by *Schistosoma haematobium* [2,3]. It is characterized by hematuria in the majority of cases. Chronic infection leads to fibrosis of the bladder and ureter, which can progress to hydronephrosis and create conditions conducive to bladder cancer. In women, urogenital schistosomiasis can cause vaginal bleeding, pain during sexual intercourse, and nodules in the vulva, now described as female genital schistosomiasis. It also causes poor cognitive development, decreased physical activity, poor academic performance, decreased productivity and work ability, as well as nutritional deficiencies and stunted growth [4].

The gold standard for the diagnosis of urinary schistosomiasis is microscopic detection via the filtration of parasite eggs present in urine [5]. However, chronic and asymptomatic infections would be difficult to detect by microscopy [6]. Indeed, it is well known that the filtration method lacks sensitivity in people with low parasite density or chronic infection due to the low number of eggs excreted in urine samples [6]. To improve the diagnosis of schistosomiasis, other diagnostic tests have been combined with microscopy, and comparative studies have also been carried out [6,7,8,9,10]. Since the early 1980s, a dipstick test for the detection of micro-hematuria has been used to diagnose *S. haematobium*. In most settings, the dipstick test gives a certain proportion of false positives, where micro-hematuria cannot be linked to *S. haematobium* [6,7]. Also, the presence of hematuria alone is not synonymous with infection by *Schistosoma haematobium* [8,9]. Likewise, immunological tests such as point-of-care circulating cathodic antigen (POC-CCA) tests have been proposed for the detection of cases of schistosomiasis. Unfortunately, these methods are more effective for the detection of *Schistosoma mansoni* than *Schistosoma haematobium* [10]. Sensitive diagnostic tools capable of detecting the presence of parasites, even at low intensities, are therefore particularly important. Therefore, molecular techniques, particularly standard and real-time PCR, have been under field evaluation for the detection of schistosomiasis cases for several years [11,12,13,14,15]. For the detection of low-intensity Schistosoma spp. infections, these techniques have proven to be more sensitive than other methods. Also, some reported cases of hybrids detected by PCR of *Schistosoma haematobium* combined with other Schistosoma species have been described using PCR [16]. Some studies have shown that PCR would, therefore, be an interesting alternative method for monitoring cases in endemic areas, in order to aim for the elimination of schistosomiasis. Also, concomitant use of urinary filtration, urine strips, and PCR could improve diagnosis and allow for better management of chronic and asymptomatic cases [13,14,15].

In Gabon, *Schistosoma haematobium* is the most common parasitic species [17,18]. Several studies have described infection with this species as moderate to low [17,18]. Also, the majority of studies carried out have used urinary filtration by microscopy as a diagnostic method. A study carried out in Lambaréné among pregnant women showed an association between hematuria and infection by *Schistosoma haematobium* [18]. Furthermore, another study on the molecular diagnosis of schistosomiasis revealed that PCR was more sensitive than microscopy, and therefore could be an alternative method for the submicroscopic detection of cases of schistosomiasis in Gabon [19]. Also, no study on the concomitant comparison of the different diagnostic methods for schistosomiasis has yet been carried out in Gabon. We hypothesize that a combination of diagnostic methods could help in the management of cases of urogenital schistosomiasis, especially asymptomatic cases. It is with this in mind that we set ourselves the objective of comparing the results of molecular analyzes by PCR to those obtained by microscopy and with urine strips, with the aim of improving the diagnosis of urinary schistosomiasis in Gabon.

## 2. Materials and Methods

### 2.1. Studies Sites

This study was carried out in the semi-rural community of Ogooué-lolo. Lastoursville and Koulamoutou are two semi-rural areas located in the South-East of Gabon, with an overall population of 9519 and 16,222 inhabitants, respectively, declared in 2016 during the last general population census [20]. Lastoursville is located at the intersection of the Ogooué River, the country’s main river, and the Transgabonais railway line and Nationale 3. The two towns are approximately 560 km (Lastoursville) and 577 (Koulamoutou) km away from Libreville, capital of Gabon, by road (Figure 1).

The river crosses the two cities and is fed by several watercourses, notably rivers, backwaters, and streams, which provide favorable conditions for the development of freshwater mollusks and allow the transmission of schistosomiasis. A study carried out about 40 years ago demonstrated the presence of *Schistosoma haematobium* in the locality [21]. Also, certain investigations carried out with healthcare personnel have revealed cases of hematuria, alluding to urogenital schistosomiasis. The majority of local populations do not have access to drinking water and resort to small tributary rivers of the Ogooué for water supply, fishing, swimming, domestic activities, and games for children. This situation is due to the fact that the municipal drinking water supply network provided by the Gabon energy and water company has an insufficient drinking water supply capacity and also to the fact that certain populations, despite the presence of a pump, prefer to frequent fresh water courses.

### 2.2. Data Collection Procedure

The investigators, including a nurse, two research engineers, a laboratory technician, and two doctoral students, received two weeks of training. During the training, the aim was to explain to them the purpose of the study, how to complete the questionnaire, and how to obtain consent from the study participants. The questions were explained in the local language for people who did not understand French. Thus, each participant was individually interviewed at the collection location. For each recruited participant, a unique identification number was assigned and information regarding age, gender, name, and place of residence was collected. A questionnaire was also submitted to each consenting adult. For minors, parents or consenting guardians were required to respond.

### 2.3. Collection of Urine Samples

After explaining the purpose of the study and obtaining written informed consent from each adult participant, parents/guardian, and child with verbal consent, urine samples were collected from people of all ages living in the study area. Study participants were then asked to provide a urine sample (midday urine) between 10:00 a.m. and 2:00 p.m. using a sterile, labeled plastic container (50 mL).

### 2.4. Detection of Hematuria

On the day of collection, all urine specimens (i.e., only one specimen per person) were tested with a urinalysis test strip (Hemastix, BAYER^®^ (Bayer Diagnostics; Basingstoke, UK)) for the presence of hematuria. After a brief immersion of the test strip in the urine sample (1 to 2 s), the test strip was removed and then left on the bench for approximately 1 min. Finally, the test strip color was compared to the corresponding reference color patches. The urinary hematuria intensity was recorded and classified as negative (0), trace (±), low (1+), medium (2+), or high (3++), according to the manufacturer’s instructions.

#### Microscopy Analysis

The urine samples were processed using the filtration technique; we used nucleopore membrane filters with a diameter of 13 mm and a pore size of 12 μm. Each urine sample was mixed well, and 10 mL was filtered through the membrane filter, which was then placed on a microscope slide. A drop of Lugol’s iodine was added, and the slide was examined under a microscope using ×10 and ×40 objectives by two experienced laboratory technicians. The number of S. haematobium eggs per 10 mL was counted and recorded for each child. 

### 2.5. DNA Extraction and PCR Tests

DNA was extracted from urine samples using the EZNA TISSUE DNA kit (Omega, Bio-tek, Norcross, GA, USA). DNA extraction, elution, and purification were carried out according to the manufacturer’s recommendations. Real-time PCR was performed first by targeting the Dra1 gene of *Schistosoma haematobium*, as described elsewhere [22]. Then, a standard PCR targeting the Cox1 and ITS2 genes of *Schistosoma haematobium*, *Schistosoma bovis*, was carried out as described here. The real-time reaction mixture was prepared in a total volume of 20 µL from 10 µL ROCHE master mix, 0.5 µL of each primer and probe, 3.5 µL pure nuclease-free water, and 5 µL matrix DNA extraction. Real-time PCR was performed on a CFX96^TM^ thermocycler (BIO-RAD, Life Science, Marnes-la-Coquette, France) for the ITS2 gene. A standard PCR test was carried out on a BIORAD brand thermocycler (BIO-RAD, Life Science, Marnes-la-Coquette, France). The reaction mixture was prepared in a total volume of 25 µL. Primers and probe sequences for real-time PCR and standard PCR are described in Table 1.

### 2.6. Data Analysis

Data were collected using a paper questionnaire designed for the study and then entered into Excel software. Statistical analyses were performed with R (version 3.4.1). Quantitative variables were described in terms of mean values, while qualitative variables were presented in terms of numbers and percentages of the data provided. For statistical comparisons, depending on the conditions of applicability, a chi-square or Fisher test was used. The test was considered statistically significant for a *p* value < 0.05. The sensitivity, specificity, positive predictive value, and negative predictive value of the test were calculated to evaluate the performance of the PCR test and the urine stick test. The level of correlation between the techniques was also assessed by calculating the kappa coefficient. It was interpreted as follows: k = 0.21–0.40, acceptable correlation; k = 0.41–0.60, moderate correlation; k = 0.61–0.80, significant correlation; and k = 0.81–1, almost perfect correlation. The Youden index was also calculated. This index allows the overall performance of a test to be assessed. This ranges from 0 to 1 and can be interpreted as follows: 0 = ineffective test, 1 = perfect test.

## 3. Results

### 3.1. Socio-Demographic Characteristics of the Population Studied

A total of 281 participants were included in this study. Among them, 122 (43.4%) were male and 159 (56.6%) were female. The average age of study participants was 21.5 ± 3.4 years, with a minimum of 2 years and a maximum of 89 years. The most represented age group was 6 to 14 years old, with a frequency of 37.7% (106/281). The majority of participants were recruited in the SETRAG district of Lastoursville (61.6%, 173/281).

### 3.2. Prevalence and Intensity of Schistosoma Haematobium Infection

Hematuria was positive in 33.5% (94/281) of the 281 participants tested. Microscopy showed that the prevalence of urinary schistosomiasis in the study area was 36.7% (103/281). The prevalence of severe infection was 20% (56/281), and that of mild infection was 16.7% (47/281). The prevalence by gender, age, and locality is presented in Table 2. The highest prevalence in microscopy was observed in children of school age (42.45%), preschool age (37.93%), and adolescents aged 15 to 17 (61.11%). In women of childbearing age, the microscopic prevalence rate was 16.3%. The microscopic prevalence was higher in Libongui 1 and SETRAG, with a prevalence of 59.65% (34/57) and 36.42% (59/173), respectively, compared to Libongui 2 (11.76%, 6/51), *p* < 0.0001.

Considering the age groups, for the Urine stick test, the highest cases of hematuria were first detected in adolescents aged 15–17 years (66.7%), then in school-aged children (39.2%) and preschool children (31.03%), and was lowest among adults aged 18–49 (30.6%). As with microscopy, the rate of hematuria was higher in Libongui 1 (59.65%, 34/57) and SETRAG (34.1%, 59/173) participants, *p* < 0.0001; Table 2.

A higher prevalence of 40.9% (115/281) was noted with the real-time PCR technique. A higher DNA load, with a Ct value < 30 (mean = 62 ± 3.2), was found in the majority of positive cases. As with microscopy, prevalence was high in school-age children (50.94%), pre-school children, and adolescents aged 15–17 years (61.11%). Likewise, the prevalence was higher in Libongui 1 (64.91%) and SETRAG (42.2%) compared to Libongui 2 (9.8%). However, except in women of childbearing age, there was an increase in prevalence with PCR in terms of age, gender, and locality compared to microscopy, although there was no statistically significant difference.

### 3.3. Threshold Range and Cycle Thresholds

Regarding the intensity of the infection, there was a good correlation between the number of eggs/10 mL of urine and the threshold range of the qPCR cycle. The average eggs/10 mL of urine was higher in the group with a Ct < 30 (211 ± 192) compared to the group with Ct values between 30 and 35 (28 ± 37.9), *p* < 0.0001. As shown in Figure 2, the range of egg counts per 10 mL of urine was wider in patients with a Ct < 30 compared to those with a Ct between 30 and 35.

### 3.4. Performance of Urine Strips and PCR Compared to Urinary Filtration

In order to evaluate the diagnostic performance of the urinalysis test strips and PCR using the Dra1 gene, the urine filtration method was used as a reference. As shown in Table 3, out of 281 specimens, 102 were positive by both urine filtration and PCR and 166 were negative by both methods. In comparison with the filtration method, the sensitivity, specificity, negative predictive value, and positive predictive value were calculated as 99.03%, 93.3%, 99.4%, and 89.5%, respectively. Only one sample was positive by filtration and negative by PCR. There was good agreement between the filtration method and the PCR (Kappa = 0.89) and excellent specificity and sensitivity with the PCR, with a good Youden coefficient (0.89).

Eighty (80) samples out of 281 were found to be positive by both urine filtration and the urine dipstick method. Furthermore, twenty-three samples were both filtration and urine dipstick-negative. Compared with the urine filtration method, the sensitivity, specificity, negative predictive value, and positive predictive value were calculated to be 74.8%, 92.1%, 87.7%, and 85%, respectively. Although these values are slightly lower than those obtained with PCR, we nevertheless noted good sensitivity and excellent specificity, with a Youden coefficient of 0.67. The level of agreement between the two diagnostic methods for detecting urogenital schistosomiasis was substantial (Kappa = 0.67).

### 3.5. Detection of Hybrid Cases

The Cox1 and ITS2 genes of *Schistosoma haematobium* and *Schistosoma bovis* were amplified in order to detect possible cases of hybridization between species. Of the 102 samples analyzed, all (100%) were positive for *Schistosoma haematobium*. No cases of hybrids were detected.

## 4. Discussion

*Schistosoma haematobium* infection continues to pose a clinically significant challenge in many developing countries [23]. In Gabon, the fight against schistosomiasis is essentially based on the massive distribution of drugs among school-age children and at undetermined frequencies, and without prior diagnosis. This study is the first to evaluate the diagnostic performance of PCR and urine strips using microscopy as the gold standard.

In our study, urinary schistosomiasis was moderate. Of the 281 participants included, 94 had hematuria (representing approximately 33.5%), and 103 were positive by microscopy (representing a prevalence of 36.7%). The moderate prevalence observed here is consistent with national trends [17,18]. This study showed that for all cases of urogenital schistosomiasis in preschool children, school-aged children, and adolescents, the number of cases was almost the same when using microscopy or urine sticks. The similarity of the urine dipstick results to microscopy in our study is consistent with observations made elsewhere [7,24]. In addition, an association between hematuria and the presence of schistosomal eggs in the urine has already been described in Gabon and elsewhere [17]. However, another study showed that detection of hematuria is low when infection levels are around 1–5 eggs per 10 mL of urine [25]. This could explain why, since the mean number of eggs was high in the different groups of our population (133.85 ± 177), the correlation between the two techniques was interesting in our study. Also, approximately 14 cases that were negative in microscopy were detected as positive in hematuria. This observation could be due to the fact that the presence of hematuria alone is not synonymous with infection by *Schistosoma haematobium* [8,9].

In some studies, the Hemastix test has been described as an interesting tool for epidemiological investigations due to its high sensitivity and low cost. However, specificity could be a limiting aspect during large-scale use [24]. However, the present study using the Urine Stick test showed not only limited sensitivity, but also excellent specificity.

Using the real-time PCR technique with the mitochondrial gene Dra1 specific to *Schistosoma haematobium* as a marker, 115 out of 281 participants were positive, which represents approximately 40.9% of cases. Although these results are not statistically different, we nevertheless note a slight increase in the number of positive cases with PCR. Also, the analysis made it possible to diagnose 13 positive cases among the negative ones in microscopy. Our results are in agreement with those observed elsewhere [24], which showed an increase in the number of cases with the PCR technique compared to microscopy, although Doudou Sow and al. showed a significant difference in prevalence between the two analysis methods [24].

Interestingly, about 16% of women of childbearing age had *Schistosoma haematobium* infection. Although we did not separate adult women from adolescents, our results corroborate those observed in Senegal among adult women, where the prevalence was around 17% [24]. School-aged children, preschoolers, and adolescents were the most affected populations regardless of the analysis method used. Our results are superior to those described by Mintsa and al. and Dejon and al. in Gabon in school and preschool-age children [17,18], as well as in studies carried out in Egypt, Senegal, and Ethiopia [23,24,26]. On the other hand, a cross-sectional study carried out by Ndassi and al. showed a decrease in schistosome prevalence in children aged 5 to 15 years [27]. This decrease could be explained by the fact that mass treatments specifically target this segment of the population. In particular, it was important to describe the prevalence among women of childbearing age, adolescents, and the rest of the population at risk, because the mass distribution of praziquantel does not often target them. In addition, evidence in the literature shows that the burden of schistosomiasis among pre-school children (PSCs) and women of reproductive age in Africa varies from country to country, and even from region to region within the same country. In West Africa, a study carried out in Burkina Faso reported a moderate prevalence of 21.3% among women of childbearing age [28], while in Tanzania, a low prevalence of around 4.5% was reported [29]. Among pre-school children, similar variations in prevalence have also been observed in some African countries [30,31,32]. These results, therefore, suggest that it would be necessary to take into account all sections of the population during control campaigns, as recommended by the WHO [1,33].

This situation is particularly interesting in the context of schistosomiasis surveillance, as the use of different diagnostic tools may alter the choice of control strategy or approach. Molecular biology could, therefore, be the most suitable diagnostic method for the detection of Schistosoma in urine samples. Indeed, the superiority of PCR techniques compared to conventional methods has been proven by several studies [14,24,34,35].

The present study showed a very good performance of PCR compared to urinary filtration by microscopy; there was excellent specificity and sensitivity. For the assessment of infection intensity, the correlation between the number of eggs per 10 mL and the Ct value was good. The average eggs per 10 mL was higher in the group with a Ct value < 30 compared to the group with a Ct value between 30 and 35. Many other studies have shown a similar good correlation between the number of eggs and the Ct value. For example, Doudou Sow et al. showed a very good correlation between the number of eggs and the Ct value [24]. Additionally, Melchers et al. also described a strong correlation [36]. In another study, the authors showed that a single schistosome egg corresponds to a specific Ct value [37]. More specifically, they found that a single schistosome egg corresponds to a Ct value of 20 ± 2. The type of DNA extraction is another important aspect of the PCR technique. Indeed, as reported by Pomari et al. [38], a DNA extraction procedure with a beating step increases the amount of Schistosoma haematobium DNA. The tandem repeat gene Dra 1 was selected for amplification in this study; this 121 bp sequence has been proposed for the detection of schistosomes at low levels of infection [7,39]. Since the Dra 1 tandem repeat is found in 15% of the total Schistosoma haematobium genome, it is used as a target gene. Although it is difficult to distinguish between DNA fragments present in urine and those extracted from eggs, the important thing is that Dra 1 fragments were detected in specimens where eggs were found and in specimens where no eggs were seen. This supports the idea that the presence of Dra 1 also appears to be independent of the number of eggs transmitted in the specimens.

Therefore, this study confirms the fact that in the context of schistosomiasis elimination, real-time PCR is a very good tool for detecting Schistosoma haematobium in urine, especially in asymptomatic and chronic cases. However, this technique, although already successful in countries endemic for schistosomiasis, requires a complex platform and expensive reagents. It is therefore important to conduct more field evaluations of isothermal amplification technologies, including LAMP and RPA, which have been shown to be more sensitive and specific than real-time PCR and less expensive [40,41].

Ultimately, the diagnosis of schistosomiasis using a combination of methods would be necessary as part of an improvement in management methods to facilitate the elimination of this disease.

The search for hybrids in this study did not make it possible to detect cases of hybrids between *Schistosoma haematobium* and *Schistosoma bovis*. This could be explained by the fact that in our study area, the populations do not practice cattle breeding; it would therefore be necessary to evaluate other cases of hybrids with other species such as *Schistosoma guineensis*, *Schistosoma mansoni*, and *Schistosoma mattei*.

## 5. Conclusions

For the optimal control and elimination of schistosomiasis as a public health problem, diagnosis is an important aspect. The WHO recommends the use of microscopic methods as the gold standard. However, more sensitive methods, notably molecular techniques, are necessary for the detection of asymptomatic and chronic cases in endemic areas. In addition, a combination of methods could be used to improve diagnosis and to diagnose the greatest number of cases. On the one hand, this study showed the importance of the real-time PCR technique, particularly in asymptomatic cases. On the other hand, it made it possible to show a good correlation between microscopy and hematuria. For public health purposes, further studies are needed to evaluate isothermal amplification technologies in the field.

## Figures and Tables

**Figure 1 diagnostics-15-01052-f001:**
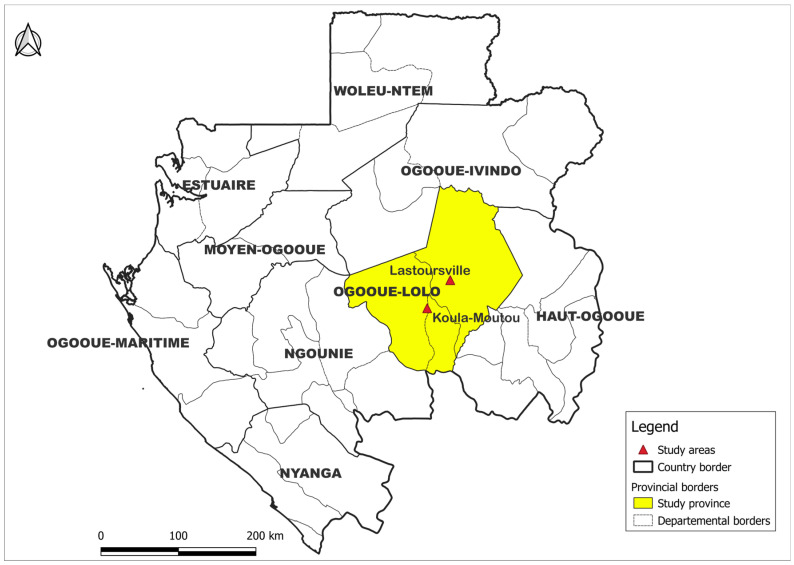
Map of Study site.

**Figure 2 diagnostics-15-01052-f002:**
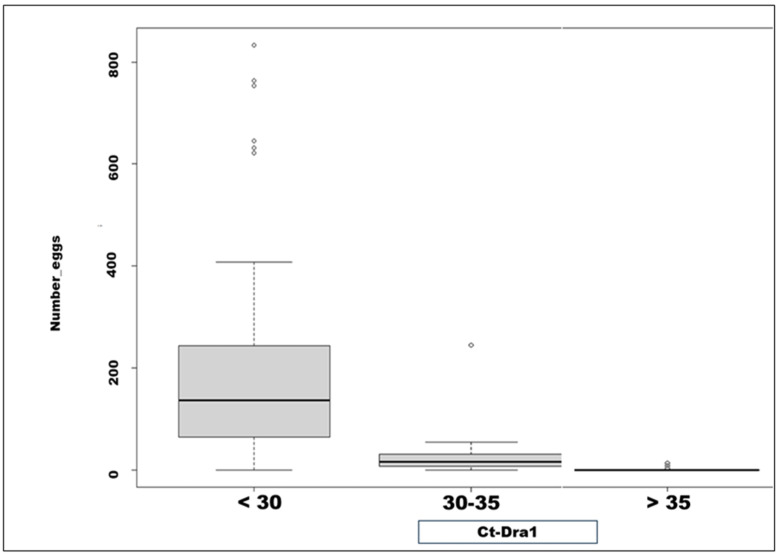
Comparison between number of eggs/10 mL and real-time PCR Ct value.

**Table 1 diagnostics-15-01052-t001:** Primers, probes, and PCR program.

Primer Names	Marker	Primer Sequences (5′-3′)	PCR Program
**Sh- FW**	DraI	GATCTCACCTATCAGACGAAAC	Denatunation: 95° to 5 min, Denaturation: 95 °C to 30 s, Annealing: 60 °C to 60 s for 39 cycles, starting from the second denaturation step.
**Sh- RV**		TCACAACGATACGACCAAC	
**Probe (FAM 5′-3′ZEN)**		TGTTGGTGGAAGTGCCTGTTTCGCAA	
**ASMIT**	COX	TTTTTTGGTCATCCTGAGGTGTAT	Denatunation: 95° to 15 min, Denaturation: 95 °C to 30 s, Annealing: 62 °C to 45 s, Elongation: 72 °C to 45 s, final elongation: 72 °C to 7 min, for 40 cycles, starting from the second denaturation step.
ShR		TGATAATCAATGACCCTGCAATAA	
**SbR**		-CACAGGATCAGACAAACGAGTACC	
**ITS2F**	ITS2	GGAAACCAATGTATGGGATTATTG	Denatunation: 95° to 55 min, Denaturation: 95 °C to 30 s, Annealing: 56 °C to 30 s, elongation:72 °C to 1 min 30 s, final elongation: 72 °C to 7 min, for 40 cycles, starting from the second denaturation step.
**ITS2R**		ATTAAGCCACGACTCGAGCA	

**Table 2 diagnostics-15-01052-t002:** Prevalence of urogenital schistosomiasis using different techniques.

Parameters	N	Hematuria		Microscopy				Real-Time PCR			
		n (%)	*p*-Value	n (%)	*p*-Value	Eggs Means ± SD	*p*-Value	n (%)	*p*-Value	CT Means ± SD	*p*-Value
**Total**	281	94 (33.5)		103 (36.7)		133.85 ± 177		115 (40.9)		28.4 ± 4.1	
**Age**			**0.0002**		**0.002**		0.8		**0.0001**		0.7
**≤5 years**	58	18 (31.03)		22 (37.93)		92 ± 132		25 (43.10)		29.8 ± 3.8	
**6–14 years**	106	42 (39.62)		45 (42.45)		135 ± 204		54 (50.94)		28.3 ± 4.2	
**15–17 years**	18	12 (66.7)		11 (61.11)		96 ± 122		11 (61.11)		28.7 ± 3.8	
**18–49 years**	62	19 (30.6)		21 (33.87)		119 ± 116		21 (33.87)		28.4 ± 3.8	
**>49 years**	37	3 (8.1)		4 (10.81)		197 ± 299		4 (10.81)		27.3 ± 5.02	
**Gender**			0.9		0.3		0.9		0.9		0.67
**Females**	159	57 (35.8)		62 (39)		103 ± 143		67 (42.1)		29 ± 3.5	
**Males**	122	37 (30.33)		41 (33.6)		143 ± 204		48 (39.34)		28.2 ± 4.6	
**Women of childbearing age**	123	20 (16.3)		20 (16.3)				19 (15.5)			
**Localities**			**<0.0001**		**<0.0001**		0.7		**<0.0001**		0.3
**LIBONGUI1**	57	34 (59.65)		34 (59.65)		131.6 ± 205.6		37 (64.91)		27.8 ± 4.2	
**SETRAG**	173	59 (34.10)		63 (36.42)		118.6 ± 158.6		73 (42.2)		29.04 ± 3.9	
**LIBONGUI2**	51	1 (2)		6 (11.76)		49.8 ± 56.7		5 (9.8)		29.5 ± 3.1	

**Table 3 diagnostics-15-01052-t003:** Performance of real-time PCR technique and test strips compared to filtration.

Real Time Pcr	Miscroscopy								
	Positive	Negative	Total	Sensitivity (95%CI)	Spécificity (95%CI)	Vpp (95%CI)	VPN (95%CI)	Kappa	Youden
Positive	102	12	114	99.03 (93–99)	93.3 (88–96)	89.5 (81–94)	99.4 (96–99)	0.89	0.93
Negative	1	166	167						
Total	103	178	281						
Hematuria									
Positive	80	14	94	74.8 (65–82)	92.1 (86–95)	85 (75–91)	87.7 (82–92)	0.78	0.67
Negative	23	164	187						
Total	103	178	281						

## Data Availability

The datasets used in this study are available upon reasonable request from the corresponding author.

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
