# Peer review of "Molecular Detection of Urogenital Schistosomiasis in Community Level in Semi-Rural Areas in South-East Gabon"

_diagnostics, 2025, doi:10.3390/diagnostics15091052_

Round 1

Reviewer 1 Report

Comments and Suggestions for Authors

New diagnostics for SCH are much needed and a priority area. This clearly written paper compares three diagnostic tools for S. haematobium – haematuria dipsticks, microscopy, and qPCR – in an endemic area in Gabon.

This is a well-written and clearly laid out paper. Below are some areas that could be addressed / strengthened before publication.

Comments

·       Lines 83-85. I’m not sure why this reference to hybrids is relevant here.

·       Line 133 – it would be useful to add who produced the urinalysis test strips

·       Methods – a description of the microscopy approach should be added

·       Table 1 is difficult to read. The column headings are crowded and it is difficult to compare the data between columns.

·       Can you add any information on what has been the history of treatment in this area? That would be useful background information and also can impact the performance of diagnostics.

·       Figure 2 needs strengthening. The axis titles should be updated and made clearer. I think the data on the x-axis need to be reordered to: <30, 30-35, >35.

·       Do you have any views on the costs / logistics / ease of use / time taken for the different methods? That will have a big impact on which is chosen of course. Urine dipsticks can be done at the point of collection, microscopy can be done with portable microscopes or in a basic laboratory. PCR needs much more sophisticated equipment. This is discussed in lines 328-332. In this case in Gabon do you think the extra expense and difficultly with qPCR worth it for the increased number of cases found? (36.7% by microscopy, 40.9% by PCR)

·       Intensity of infection remains an important metric. With the qPCR, is there a number of Ct that you would suggest equates to heavy infection?

·       Lines 331-333: mention LAMP and RPA diagnostic techniques. I think these should be mentioned earlier as well, in the introduction.

 Minor / Editorial

·       The English is generally of a very high standard. There are a couple of small errors that an be updated, e.g: Line 95: ‘semi-rule’ should probably be ‘semi-rural’, Line 138: ‘large’ should probably be ‘high’, Line 197: ‘children aged school’ should probably be ‘school-aged children’

·       Please add a legend to Table 2 explaining acronyms

Author Response

Reviewer 1:

Reviewer comments:

New diagnostics for SCH are much needed and a priority area. This clearly written paper compares three diagnostic tools for S. haematobium – haematuria dipsticks, microscopy, and qPCR – in an endemic area in Gabon.

This is a well-written and clearly laid out paper. Below are some areas that could be addressed / strengthened before publica

Responses to reviewer

Introduction

 1)-  Lines 83-85. I’m not sure why this reference to hybrids is relevant here.

Right! Based on these good, We have deleted this reference

Method

2) Line 133 – it would be useful to add who produced the urinalysis test strips

Right! (Hema-Combistix®, Siemens),  We have added the product reference.

 Methods – a description of the microscopy approach should be added

Right! We have added the microscopy methods used.

The urine samples were processed using the filtration technique, we used nucleopore membrane filters with a diameter of 13 mm and a pore size of 12 μm. Each urine sample was mixed well and 10 ml was filtered through the membrane filter which was then placed on a microscope slide. A drop of Lugol's iodine was added and the slide was examined under a microscope using ×10 and ×40 objectives by two experienced laboratory technicians. The number of S. haematobium eggs per 10 ml was counted and recorded for each child. In method section line143-149

3- Table 1 is difficult to read. The column headings are crowded and it is difficult to compare the data between columns.

We have reorganised table 1, enlarging it slightly to make it easier to read.

  • Can you add any information on what has been the history of treatment in this area? That would be useful background information and also can impact the performance of diagnostics.

Thank you for your pertinent comment. We would have liked to include data on treatment in Gabon, but we have no data available on the treatment policy, so diagnostic campaigns are carried out sporadically.

4- Figure 2 needs strengthening. The axis titles should be updated and made clearer. I think the data on the x-axis need to be reordered to: <30, 30-35, >35

Right! We have revised the figure and the new figure is presented as follows

  • 5- Do you have any views on the costs / logistics / ease of use / time taken for the different methods? That will have a big impact on which is chosen of course. Urine dipsticks can be done at the point of collection, microscopy can be done with portable microscopes or in a basic laboratory. PCR needs much more sophisticated equipment. This is discussed in lines 328-332. In this case in Gabon do you think the extra expense and difficultly with qPCR worth it for the increased number of cases found? (36.7% by microscopy, 40.9% by PCR)

It is well known that PCR is more expensive than other diagnostic methods and that microscopy is the gold standard for the detection of schistosomiasis eggs. However, if we want to achieve elimination, PCR should be used for epidemiological studies and not for routine diagnosis. So, for Gabon, PCR would be very welcome for epidemiological studies.

  • 6- Seventy-five percent of the Ct-values were ≥ 33 in the egg-negative category, < 31 in the light intensity category, and < 24 in the heavy intensity category.

Giving an exact number of eggs would be difficult to say, it is certainly a question to be explored, however, some studies have shown that the higher the intensity of infection, the lower the Ct-value.

  • The English is generally of a very high standard. There are a couple of small errors that an be updated, e.g: Line 95: ‘semi-rule’ should probably be ‘semi-rural’, Line 138 (140): ‘large’ should probably be ‘high’, Line 197 (230): ‘children aged school’ should probably be ‘school-aged children’

Right! We have made corrections to these spelling errors.

Reviewer 2 Report

Comments and Suggestions for Authors

After reading the text of the article, the importance and novelty of the work "Molecular detection of urogenital schistosomiasis in community level in semi-rural areas in South-Est Gabon" does not cause me any doubts. The text is articulated in clear scientific language, and the research methods and statistical analyses appear to be well-chosen. However, I have a few comments that I believe could enhance the clarity and quality of the manuscript:

1. The "Materials" section is currently missing and should be included for completeness.

2. I find the "Study Sites" section somewhat redundant given the focus of the study and the article's title. It may be more appropriate to relocate this information to the accompanying materials.

3. In lines 128-130, authors mention that participants were "asked to provide a urine sample (midday urine) between 10:00 a.m. and 2:00 p.m." Please, clarify why this specific time frame was chosen? If these are general recommendations, please provide a reference to support this.

4. Tables 1 and 2 are difficult to read. I recommend aligning all text within these tables for improved readability.

5. In lines 133-134, authors state, "After briefly dipping the test strip into the urine sample (1-2 seconds)."Please, specify why this duration is recommended? If it is based on the manufacturer's guidelines, please include a reference. Additionally, the manufacturer of the test strip should be identified.

Author Response

Reviewer 2

  1. The "Materials" section is currently missing and should be included for completeness.

We take your comment into account, and we have included this part in the method section and changed the title to Materials and Methods. Materials and Methods Section line 95

  1. I find the "Study Sites" section somewhat redundant given the focus of the study and the article's title. It may be more appropriate to relocate this information to the accompanying materials.

Right! Nous comprenons votre préoccupation, cependant, cette partie est importante vu qu'elle permet au lecteur de mieux s'orienter et localiser la zone d'étude et le faciès socio-démographique.

  1. In lines 128-130, authors mention that participants were "asked to provide a urine sample (midday urine) between 10:00 a.m. and 2:00 p.m." Please, clarify why this specific time frame was chosen? If these are general recommendations, please provide a reference to support this.

Thanks for this observation. For the performance of urine tests for the search for schistosoma haematobium, it is recommended to do the test after physical effort, the times are not necessarily strict, these are the times at which we were led to go to the field for the collection, it was also not necessary to keep the eggs too long in the heat to avoid hatching

  1. Tables 1 and 2 are difficult to read. I recommend aligning all text within these tables for improved readability.

Right! We have aligned the text and enlarged the table slightly for good readability

  1. In lines 133-134, authors state, "After briefly dipping the test strip into the urine sample (1-2 seconds)."Please, specify why this duration is recommended? If it is based on the manufacturer's guidelines, please include a reference. Additionally, the manufacturer of the test strip should be identified.

This duration is not recommended, this is our analysis method. But the reading of the urine test should not exceed 2 minutes in general. The urinary strip used was / Hemastix, BAYER® . Materials and method section line 136.